# Health and Well-Being in Protected Natural Areas—Visitors’ Satisfaction in Three Different Protected Natural Area Categories in Catalonia, Spain

**DOI:** 10.3390/ijerph17186746

**Published:** 2020-09-16

**Authors:** Estela Inés Farías-Torbidoni, Sebastià Mas-Alòs, Gonzalo Gil-Moreno-de-Mora, Pere Lavega-Burgués, Marta Castañer, Eloisa Lorente-Catalán, Jordi Seguí-Urbaneja, Enric Lacasa-Claver

**Affiliations:** 1Institut Nacional d’Educació Física de Catalunya (INEFC), Universidad de Lleida (UdL), 25192 Lleida, Spain; smas@inefc.es (S.M.-A.); ggil@inefc.es (G.G.-M.-d.-M.); plavega@inefc.udl.cat (P.L.-B.); castaner@inefc.udl.cat (M.C.); elorente@inefc.udl.cat (E.L.-C.); jsegui@inefc.es (J.S.-U.); qlacasa@inefc.es (E.L.-C.); 2Grup d’Investigació Social i Educativa de l’Activitat Física i de l’Esport (GISEAFE), INEFC, Universidad de Barcelona (UB), 08038 Barcelona, Spain; 3Grup de Recerca Moviment Humà, Universidad de Lleida (UdL), 25192 Lleida, Spain; 4Grup de Recerca en Desenvolupament i Innovació de la Condició Física i l’Exercici (DICFE), INEFC, Universidad de Lleida (UdL), 25192 Lleida, Spain; 5Grup d’Investigació en Acció Motriu (GIAM), INEFC, Universidad de Lleida, 25192 Lleida, Spain; 6Grup d’Observació de la Motricitat i l’Esport, Innovació en Dissenys Observacionals i Mixed Methods Research, INEFC, Universidad de Lleida (UdL), 25192 Lleida, Spain; 7Grup d’Investigació sobre l’Activitat Física i la seva Didàctica per a l’Educació, la Cultura i el Benestar (DECUBIAF), INEFC, Universidad de Lleida (UdL), 25192 Lleida, Spain; 8Grup de Recerca en Sistemes Complexos i Esport, INEFC, Universidad de Lleida (UdL), 25192 Lleida, Spain

**Keywords:** physical activity, health-related physical activities, motivations, benefits, national parks, natural parks, periurban parks, management

## Abstract

Protected natural areas (PNAs) can be a source of health and well-being, but little research has been carried out regarding outcomes in terms of satisfaction (the difference between motivations and benefits). Inspired by previous research that examines the motivations and benefits perceived by visitors to various PNAs in Canada, and based on importance–performance analysis (IPA) and service quality gap (GAP) analysis theory, the aim of this study was to identify the outcomes generated by protected areas in terms of satisfaction, especially with regard to the PNAs’ individual protection categories. The study was based on survey data from visitors (*n* = 360) to three PNAs in Catalonia: one national park, one natural park and one periurban park. The results indicate that anticipated environmental, psychological, physical and social benefits were of major personal value in choosing to visit a PNA. The results indicate that, generally, visitors were satisfied with regard to the benefits anticipated. Differences between parks in this respect could be explained in part by sociodemographic factors and visitors’ behavior. The results are discussed in terms of their applicability and how they relate to the role of PNAs in the promotion of visitors’ health and well-being.

## 1. Introduction

At the global level, visits to protected natural areas (PNAs) have been increasing since the 1990s, reaching unprecedented numbers [1]. According to Europarc–Spain’s statistics, 15 million people visited Spanish national parks in 2017, in contrast to only 8 million in 2007 [2]. In the case of Catalonia, the increase in numbers is similar: in 2018, more than 4.2 million people visited one or more of the 14 Catalonia natural parks, 1 million more than in 2008 [2].

Although the functions of PNAs were originally limited to aesthetic and protective ones, new roles and functions of protected areas (PAs) have developed over time [3,4]. One recently acknowledged new role has been to see PAs as prominent legal arenas for sustaining active lifestyles, with a wide range of nonmaterial benefits such as improving health and well-being through participation in a variety of health-related physical activities (HEPA) [5,6,7,8,9]. The management of PNAs may address broader policies and strategies to promote health by influencing individual lifestyle factors (e.g., increasing physical activity levels) or promoting social networks (e.g., engaging people in outdoor group activities) [10]. More precisely, physical inactivity was identified as the tenth most important global risk factor for chronic diseases [11] and the fourth leading risk factor for global mortality [12]. More recently, the Bangkok Declaration on Physical Activity for Global Health and Sustainable Development identified strong links between HEPA promotion strategies and the 2030 Agenda for Sustainable Development Goals (SDGs). Three of these goals can be addressed by PNA management: SDG3—Ensure healthy lives and promote well-being; SDG11—Inclusive, safe, resilient and sustainable cities and communities; SDG15—Life on Land [13]. In 2015, with the 2030 Agenda, the United Nations General Assembly committed to ensuring increased global health coverage and reducing health inequities via policy actions aimed at increasing physical activity levels [14]. Action 2.4 of the Agenda is to “Strengthen access to good-quality public and green open spaces …” [14] (p. 33); Action 3.3 is to “Enhance provision of, and opportunities for, more physical activity programmes and promotion in parks and other natural environments…” [14] (p. 37).

To date, the principal lines of research linking healthy lifestyle, well-being and HEPA participation may be summarized as follows: (1) assessing the benefits derived from contact with natural areas and green spaces [15,16,17,18,19,20,21], (2) self-perceived personal well-being through visiting PNAs [22,23,24,25,26,27] and (3) small-scale studies examining the level of benefits and the contribution to the amount of HEPA derived from visiting PNAs [7,28,29,30]. One use of quantitative data to monitor HEPA in relation to physical inactivity as a risk factor is to analyze the physical activity of PNA visitors in terms of their metabolic equivalent (MET) consumption. Visitors’ physical activity behaviors can be categorized by intensity (how strenuous the physical activity is; MET-min) or volume (how much physical activity; MET-h) [31]. Both categorizations show correlations with other health outcomes in all ages and for most chronic health conditions [12,32].

One of the first studies to address health outcomes in PNAs was developed by Lemieux et al. [22], who considered ten different dimensions of health and well-being (physical, psychological, social, intellectual, spiritual, ecological, environmental, cultural, occupational and economic), and found that visitors reported more benefits in terms of their psychological, social, physical and environmental health. Recently, three similar studies in Europe, each of which used a different approach, reached similar results [24,26,27]. These studies analyzed the differences among visitors’ sociological characteristics (e.g., age, gender, educational background), visitor behaviour and even the specific PNA visited. Another study [26] showed important differences between visitor ratings according to the areas studied, especially in relation to social and physical well-being. In their recent study, Lemieux et al. [23] point to the importance, for planning, of taking into consideration the distinct ecosystems and roles played by different PNAs in providing health and well-being benefits.

Despite the growing interest in visitors’ perceptions of PNAs’ contributions to health and well-being, specific studies that examine visitors’ actual satisfaction levels with this role are missing. Knowing the level of park visitors’ satisfaction in relation to their health motivation and the benefits they derive from such visits could help PNA managers to develop effective management strategies to increase health and well-being benefits.

According to Arabatzis et al. [33], PNA visitors’ satisfaction can be indirectly measured through the stimuli they receive when they come into contact with the natural characteristics that create the individuality of a particular PNA. Basically, for these authors, the evaluation of visitor satisfaction depends on comparing visitors’ expectations before visiting the PNA, and the experiences and images they come into contact with.

In the context of measuring the quality of services as perceived by visitors or visitor satisfaction in a tourist destination, including a number of PNAs, several approaches were developed. These approaches were based on two main types of analysis: importance–performance analysis (IPA) and gap analysis of service quality (GAP) [33,34,35,36,37,38,39]. The first approach combines a measure of performance with its associated importance in a two-dimensional grid. It provides a graphic representation of the performance of managers, suppliers or operators in providing a range of services. The second approach measures service performance as the difference between expected performance and an evaluation of the services as actually experienced. It is important to highlight the research carried out by Tonge and Moore [38] and Rice et al. [39], who reconceptualized IPA and GAP first to assess satisfaction [38], and more recently to examine motivation outcomes [39]. Tonge and Moore [40] replaced the performance element of IPA with satisfaction, and the GAP relied on the (statistical) means of importance and satisfaction. For Rice et al. [39], on the other hand, motivation is distinct from expectation—i.e., one’s perceived likelihood of attaining the outcomes or benefits sought through a recreation experience. A negative statistically significant gap, where the importance/motivation mean is larger than the satisfaction/outcomes mean, suggests that management action is required. Conversely, a positive statistically significant gap, where the importance/motivation mean is lower than the satisfaction/outcomes mean, suggests no extra management measures are required.

In our framework, which takes these new approaches into account, the original IPA (importance–performance analysis) is conceptualized as Motivation and Benefits (see Figure 1). For instance, the Importance may be to relieve stress or to do physical activity. If successful, the benefit of the recreational experience would be the attainment of a relaxed state or to have an active visit.

The aim of the present study was to compare visitor satisfaction in terms of health and well-being motivations and perceived benefits at three different PNAs (one national park, one natural park and one periurban park), in order to identify any differences between different types of PNAs in their capacity to contribute to visitors’ health and well-being. The key research questions addressed in this study follow:

(Q1) To what extent did visitors to different categories of PNA differ in terms of their sociodemographics and visit behaviors?

(Q2) What were the most important health and well-being motivations and benefits (outcomes) identified by visitors?

(Q3) To what extent did motivations and benefits differ among PNAs?

(Q4) To what extent did each individual PNA satisfy the health and well-being motivations pursued by visitors (i.e., GAP analysis)?

(Q5) For each PNA, were there any differences in the gaps between satisfaction and motivation that can be explained by sociodemographic characteristics and visit behaviors?

## 2. Materials and Methods

### 2.1. Study Areas

The study was carried out in three different PNAs in Catalonia: one national park, one natural park and one periurban park. Figure 2 shows their geographical locations: Aigüestortes i Estany de Sant Maurici National Park (A), Alt Pirineu Natural Park (P), and Serra de Collserola Periurban Park (C). The main reasons for choosing these three PNAs were that (1) all are Natura 2000 sites, (2) all are managed by public organizations, and (3) although they all provide similar opportunities in terms of recreational activities, they present different physical (i.e., topographical), social and managerial settings.

According to the recreation opportunity spectrum proposed by Brown et al. [40], Aigüestortes i Estany de Sant Maurici National Park is semiprimitive nonmotorized (SPNM); Alt Pirineu Natural Park is semiprimitive motorized (SPM), and Serra de Collserola Periurban Park is urban (U).

Aigüestortes i Estany de Sant Maurici National Park, in the Pyrenees, was established in 1955 and extended in 1996, and is the only national park in Catalonia. Alt Pirineu Natural Park, which is also in the Pyrenees, is the largest natural park in Catalonia, with an area of over 79,300 ha. Serra de Collserola, although a natural park, is one of the largest periurban parks in Europe. It receives around 5,000,000 visits per year (Table 1).

One of the most important characteristics of Aigüestortes i Estany de Sant Maurici National Park is its relief—u-shaped valleys, glacial cirques, ponds, gorges and screes—a legacy of the erosive action of Quaternary glaciers, mostly on granite, but also slates, shales and other rocks. Alt Pirineu Natural Park, located near the Aigüestortes i Estany de Sant Maurici National Park, was established in 2003 and extended in 2018. It includes the highest peak of the Catalan Pyrenees and is one of the most popular hiking areas in the Pyrenees, suitable for both day-trippers and overnight hikers. Finally, Serra de Collserola Natural Park is located in the middle of one of the densest urban areas on the Mediterranean shore, Barcelona (1.6 million residents). It was established in 1987 by the Catalan government, covers an area of 8259 hectares (17 km long × 6 km wide) and is characterized by an extensive network of recreational trails that are used for multiple activities, including hiking, mountain biking and running.

### 2.2. Data Collection and Sampling Strategy

Data were collected using 360 structured questionnaires completed onsite, 120 in each PNA. Regarding sample strategy, the interviews were held from April to July 2016 and took place at popular locations in the parks (the main entrance, the main hut and the main landscape attraction point). The questionnaires were conducted by 21 trained research staff. Potential respondents were approached randomly at the main entrances on their way out, because most of the questions referred to the experience that the visitors had just had, during their visit. The average time spent on the interview was 8–10 min and participants did not receive any financial compensation for participating in the study. The response rate was 96%.

The final sample showed similar sociodemographic data to other visitors studied in previous research by Farías and Morera in Serra de Collserola Periurban Park [41] and Alt Pirineu Natural Park [42]. Sociodemographic outcomes such as gender and age among this and previous studies were below 7% of difference (1.5% and 2.7% for gender and age, respectively, for Serra de Collserola Periurban Park and 6% and 6.6% for Alt Pirineu Natural Park). Complete data about the results obtained are shown in Section 3.

### 2.3. Questionnaire

The survey was in three parts. First, visitors were asked about sociodemographic characteristics (e.g., age, place of residence). In the second part, they were asked about recreational behavior (e.g., frequency of visit, duration of their stay, main activity practiced during the visit). Finally, visitors were asked to rate the importance of each of six motivations and their satisfaction level or outcomes for each of six possible benefits, as suggested by Lemieux et al. [22].

The ratings for motivations used a 5-point Likert scale ranging from 1 (not at all important) to 5 (very important). The motivations were physical (the opportunity to do physical activity), psychological (the opportunity to relax, or to recover from stress or mental fatigue), social (the opportunity for increased social interaction and to meet new people), spiritual (the opportunity to connect with nature or look for inspiration), environmental (the opportunity to be outdoors, connect to nature or experience a sense of place) and intellectual (the opportunity to engage in creative and stimulating activities, or to acquire new knowledge).

Where benefits were concerned, visitors were asked to what extent they thought that the visit had increased their physical, psychological, social, spiritual, environmental and intellectual well-being (from 1 = did not improve at all, to 5 = improved greatly). The benefits were defined as follows: physical (perceived physical well-being), psychological (recovery from mental stress, improved mood), social (strengthened social relations, meeting new people), spiritual (finding inspiration, connecting with nature), environmental (enjoying being outdoors, experiencing a sense of place, fostering a harmonious human–nature relationship) and intellectual (learning something new, enjoying participating in creative activities).

### 2.4. Data Analysis

The data collected were analyzed using the Statistical Package for the Social Sciences 18.0 (SPSS, Chicago, IL, USA). First, preliminary analyses were carried out, including the identification of missing values and the distribution of the data, and descriptive statistics for each of the study’s variables were defined. Secondly, we tested for possible differences between sociodemographic characteristics and visit behavior in the three PNAs. Chi-square goodness-of-fit at a 95% accuracy level was used to examine the differences between visitors with respect to categorical variables. Thirdly, since the data were not normally distributed, nonparametric Kruskal–Wallis and Mann–Whitney U tests were performed to explore the differences in motivations and benefits between visitors to the three different PNAs. Differences between the three parks were analyzed using Kruskal–Wallis tests, and differences between pairs of parks using Mann–Whitney tests. In the latter analysis, the level of significance was adjusted to *p* < 0.017 (0.05/3). Fourthly, in order to test the differences between visitors’ motivations in visiting the PNA and their satisfaction after their visit, we calculated the gap value for each motivation and benefit. Gaps were estimated by subtracting the mean for motivation (importance) from the mean for benefits (satisfaction). Six gaps were calculated for each PNA, each of them referring to a different motivation/satisfaction pair (e.g., psychological). A negative gap (motivation exceeding benefits) suggests that satisfaction is negative and additional management action is required. Conversely, a positive gap (benefits exceeding motivation) suggests that satisfaction is positive. To test whether the gaps differed between PNAs, we also used Kruskal–Wallis and Mann–Whitney tests, following the same rationale as the one described above. Finally, for each PNA we tested whether characteristics of visit behavior were related to differences in the gaps between satisfaction and motivation. Kruskal–Wallis and Mann–Whitney tests were also used in this analysis.

## 3. Results

### 3.1. Visitor Profiles and Recreational Behavior

As shown in Table 2, the most frequent visitors to the three areas were men (58%), middle-aged (40.4%), and held a university degree (57.5%). The educational level of the sample in the study was 10% higher than the average level for the Catalan population as a whole (last available census, 2017) [43]. Over two-thirds of the sample had visited the parks at least four times in the previous two years. People who were visiting the parks as part of a group most commonly came with a group of friends (43%); mostly spent about half a day in the park (78.3%); and during their visit, 60.5% undertook recreational hiking (slow walking) or hiking (brisk walking).

Regarding differences in sociodemographic characteristics and recreational or visit behavior between parks (Table 2), the results revealed significant differences in gender (*p* = 0.033), age (*p* = 0.021), frequency of visit (*p* < 0.001), group composition *p* < 0.001), duration of visit (*p* < 0.001) and recreational activity (*p* < 0.001). No significant differences were found concerning educational level (*p* = 0.094) and health perception (*p* = 0.286).

Specifically, visitors to Aigüestortes i Estany de Sant Maurici National Park were often women (50.8%) and older adults (41.6% were over 55), who were visiting the area for the first or second time (44.5%), in couples or family groups (57.5%). Almost 51% did slow walking (recreational hiking). 

Visitors to the Serra de Collserola Periurban Park were characterized by being male (65.5%) and/or young (almost the 20% were aged 26–35), visit the area frequently (73.4%), came with a group of friends (40.8%), for half a day (99.2%). They carried out more intense physical activity than the visitors to the other two parks: more than 47% of them did trail running or mountain biking during their visit. Significantly, the second most common “group” composition was visitors who were there alone (26.7%), a much higher figure than for the other two PNAs.

Finally, visitors to the Alt Pirineu Natural Park, who had an intermediate profile, were mostly male (58.8%) and middle-aged (54.7%), and 50% accompanied by a group of friends. They also remained in the area longer than visitors in the other two parks (45%) and performed less physical activity. More than the 33% of the visitors to this park remained in or near the entrances, meaning that they did not practice any type of physical activity during their visit.

### 3.2. Health and Well-Being Motivation for Visiting Protected Natural Areas

As shown in Table 3, higher health and well-being motivations to visit the PNAs (both motivational factors take into consideration the mean value and the sum of the percentages for answers corresponding to values 4 and 5—i.e., the sum column) related to the following considerations (in descending order of importance): environmental (89.4%, M = 4.52, SD ± 0.869), psychological (82.3%, M = 4.31, SD ± 1.005) and physical (76.3%, M = 4.15, SD ± 1.097), social (71.1%, M = 3.90, SD ± 1.407), spiritual (63,1%, M = 3.66, SD ± 1.367). The least important motivation was associated with intellectual considerations (38.1%, M = 2.98, SD ± 1.364). 

### 3.3. Perceived Health and Well-Being Benefits Derived from Visiting Protected Natural Areas

For benefits, the highest health and well-being scores related to the following considerations, in descending order: environmental (91.1%, M = 4.57, SD ± 0.825), psychological (91.1%, M = 4.51, SD ± 0.828) and physical (83.3%, M = 4.31, SD ± 1.015), social 78.9%, (M = 4.14, SD ± 1.312), spiritual (64.7%, M = 2.97, SD ± 1.405) and intellectual motivations (38.3%, M = 2.97, SD ± 1.405). In all cases, the perceived benefits (outcomes) equal or exceed the average motivation scores shown in Table 3 and match closely the motivations to visit the parks (Table 4). 

### 3.4. Motivation and Benefits Differences among Parks 

As shown in Table 5, the results of Kruskal–Wallis tests revealed that the parks differed significantly for three of the six motivations relating to health and well-being. Interesting differences were observed for physical, social and intellectual motivations. Visitors to Serra de Collserola Periurban Park showed higher scores for physical motivation than those to Aigüestortes i Estany de Sant Maurici National Park and Alt Pirineu Natural Park (*p* < 0.001). Similarly, visitors to Aigüestortes i Estany de Sant Maurici National Park and Alt Pirineu Natural Park exhibited higher scores for social motivation compared to visitors to Serra de Collserola Periurban Park (*p* < 0.001). Furthermore, visitors to Aigüestortes i Estany Sant Maurici National Park presented higher scores for intellectual motivation than visitors to Alt Pirineu Natural Park and Serra de Collserola Periurban Park (*p* < 0.001). No statistically significant differences were observed for psychological, spiritual and environmental motivations, two of the three most important motivations reported in the set of the parks (environmental and psychological).

Significant differences between parks were identified for four of the six underlying benefits: physical, social, environmental and intellectual (Table 6). Visitors to Aigüestortes i Estany de Sant Maurici National Park showed higher scores for environmental (*p* = 0.002) and intellectual (*p* < 0.001) benefits than those to Serra de Collserola Periurban Park and Alt Pirineu Natural Park. Likewise, Alt Pirineu Natural Park presented higher scores for social benefits (*p* < 0.001) and Serra de Collserola Periurban Park for physical benefits (*p* < 0.001).

### 3.5. Health and Well-Being Satisfaction Pursued by Visitors to Different PNAs

This subsection presents the differences (i.e., gaps) between PNAs for the six health and well-being dimensions. For the former, the results (see Table 7) revealed that the calculated gaps differed across parks only for the physical and spiritual dimensions. The two highest values of these dimensions were registered in Aigüestortes i Estany de Sant Maurici National Park, while the lowest values for these two dimensions were identified in Alt Pirineu Natural Park. However, a positive gap (higher mean for benefits than for motivation) was observed across all three parks for these same two dimensions, as well as for the psychological and physical dimensions. When further analyses were carried out, no negative gap was found in Aigüestortes i Estany de Sant Maurici National Park, while several were found in the other two PNAs—two in the case of Alt Pirineu Natural Park (environmental and intellectual), and one in Serra de Collserola Periurban Park (intellectual). Of these, only one, according to IPA theory—the intellectual dimension in Serra de Collserola Periurban Park (the value for which was outside acceptable margins)—would require intervention by management (Figure 3).

### 3.6. Influence of Sociodemographics and Recreation Behaviors on GAP Results

When examined in relation to sociodemographic and visit behavior, the gap values did not vary by age or educational level. However, several gap differences were obtained for the remaining independent variables (i.e., gender, health perception, frequency of visit, visitor group, duration of visit, and activity; see Table 8). Overall, eleven significant differences were found in Aigüestortes i Estany de Sant Maurici National Park (i.e., gender, frequency of visit, group composition, and activity); one in Alt Pirineu Natural Park (i.e., duration of visit), and four in Serra de Collserola Periurban Park (i.e., group composition, duration of visit, and activity). Subsequent post hoc analysis found six significant differences in Aigüestortes i Estany de Sant Maurici National Park, one in Alt Pirineu Natural Park, and two in Serra de Collserola Periurban Park, as the *p* value was corrected for multiple comparisons (see Appendix A). In the case of Aigüestortes i Estany de Sant Maurici National Park, differences were found according to gender (M-values: men = 0.189 vs. women = 0.079; *p* < 0.417; *p* < 0.001), heath perception (M-values: good = 0.0439 vs. very good or excellent = 0.0147; *p* < 0.013 and poor or fair = 0.549 vs. very good or excellent = 0.038; *p* < 0.044, depending on dimension), frequency of visit (M-values: first time = 0.652 vs. more than four times = 0.079; *p* < 0.001), and activity (M-values: recreational hiking = 0.433 vs. hiking = 0.116, *p* = 0.003). Thus, women who are visiting the park for the first time, reporting good health perception and doing recreational hiking during their visit are more satisfied with the physical dimension. Significant differences were also found in the environmental, social and spiritual dimensions for recreational hiking (M = 0.100) and trail running (M = 0.200, *p* = 0.006), hiking (M = 0.069) and trail running (M = 0.200, *p* = 0.004), men and women (i.e., M-values: men = 0.034 vs. women = 0.400; *p* < 0.416; *p* < 0.001) and level of health perception (M-values: poor or fair = 0.549 vs. very good or excellent = 0.038; *p* < 0.044). These findings show a greater level of satisfaction in the environmental dimension for those visitors who practice trail running during their visit, in the social dimension for visitors whose health perception was poor or fair and in the spiritual dimension for women. For Alt Pirineu Natural Park, results exhibited a significant difference in the psychological dimension regarding the duration of the visit (M-values: half a day = 0.394 vs. one day = −0.132, *p* = 0.010), showing a greater level of satisfaction in the psychological dimension for people visiting the park for half a day. Finally, results for Serra de Collserola Periurban Park revealed differences in the social dimension regarding group composition (M-values: individual = −0.125 vs. group of friends = 0.611, *p* < 0.001) and in the spiritual dimension according to the activity (M-values: mountain bike = −346 vs. trail running = 0.323, *p* = 0.001). Thus, visitors to Serra de Collserola Periurban Park reported a greater level of satisfaction in the social dimension when they visited the PNA with their friends, and a higher level of satisfaction with the spiritual dimension when they practiced trail running. 

## 4. Discussion

Using three different PNAs as case studies, this research set out with the aim of answering five questions. The results obtained provide an interesting foundation for detailed discussion.


**(Q1) To what extent did visitors to different categories of PNA differ in terms of their sociodemographics and visit behaviors?**


The results obtained revealed certain patterns of characteristics and recreational behavior in common for all three parks, for all variables with the exception of educational level. However, each park presented significant differences in terms of visitor profiles. According to earlier studies (e.g., by Farías-Torbidoni; Baric, Anic and Bedoya; Múgica and Delucio or Delucio and Múgica; [44,45,46,47,48]), Aigüestortes i Estany de Sant Maurici National Park could be characterized as attracting visitors with a more general profile than the other two PNAs, and Serra de Collserola Periurban Park as being visited by people with a more sports-oriented profile. Here, it is important to highlight that Aigüestortes i Estany de Sant Maurici National Park and Serra de Collserola Periurban Park are situated at opposite ends of the recreational opportunity spectrum (referred to in Section 2): semiprimitive nonmotorized in the case of the former and urban in the case of the latter.

The results support the outcomes found previously regarding the influence of PNA characteristics (physical, social and managerial) not only on visitor profiles, but also on the main recreational activities carried out in the PNAs. As mentioned in Section 3 (Results), while visitors to Aigüestortes i Estany de Sant Maurici National Park were often first- or second-time visitors, in couples or families, there for half a day and doing recreational hiking (slow walking), visitors to Serra de Collserola Periurban Park tended to visit frequently, individually and to carry out higher levels of physical activity (moderate to vigorous). 


**(Q2) What were the most important health and well-being motivations and benefits (outcomes) identified by visitors?**


The results obtained for main motivations and benefits were similar to those of earlier studies carried out in Canada [23,32], Finland [24] and, indeed, Spain [27]. It is worth underlining this coincidence with other studies regarding the four main motivations and benefits identified, although their order of significance may differ, for instance, physiological, social, physical and environmental aspects in the case of Canada, and environmental, psychological, physical and social in our case. Although the results are consistent, when we assess the differences between studies one limitation emerges, which is connected to the possible misunderstanding of some items. For example, while physical, psychological and social motivations and benefits could be clearly identified by users, this was not the case for environmental, spiritual and intellectual motivations or benefits. It is possible that respondents found it difficult to identify and interpret aspects such as connecting with nature or enjoying the outdoors (environmental), seeking inspiration (spiritual) or having the opportunity to participate in creative and stimulating activities in order to acquire new knowledge (intellectual). Thus, it is possible that not all the answers reflect the same interpretations.


**(Q3) To what extent did motivations and benefits differ among PNAs?**


The results obtained regarding this question not only support the existence of significant differences between parks, but also allow us to identify some specific well-being roles (related to the identity and character of each PNA) attributed by visitors to the parks. Although there is a perfect match between the three highest values for motivations and the perceived benefits (outcomes) of visiting each park, there are significant differences in the specific motivations and benefits associated with each park. For instance, environmental and intellectual benefits are clearly associated with Aigüestortes i Estany de Sant Maurici National Park, social benefits with Alt Pirineu Natural Park, and physical benefits with Serra de Collserola Periurban Park. However, these trends need to be carefully considered because the research did not go into depth with all possible associations; for instance, Serra de Collserola Periurban Park is visited weekly by visitors and their results show low scores for the intellectual dimension, which may be related with the high frequency of visits. These results are especially significant if we associate them with the visitor profiles identified by Q1. Furthermore, the results obtained in the case of Serra de Collserola Periurban Park (a more sports-oriented profile, with greater importance attached to physical motivation and benefits) again support the important role that this kind of area can play in the promotion of HEPA [14,28,31,49,50].


**(Q4) To what extent did each individual PNA satisfy the health and well-being motivations pursued by visitors (i.e., Gap analysis)?**


The data obtained show that generally, in this set of PNAs, there is a concordance between what visitors want (motivation) and what they get (benefits). In the case of these three parks, only three gap values emerged (i.e., where benefits derived by visitors did not meet with their motivations): the environmental and intellectual dimensions in Alt Pirineu Natural Park, and the intellectual dimension in Serra de Collserola Periurban Park. From IPA and GAP points of view, the intellectual dimension in Serra de Collserola Periurban Park is the only dimension in any of the three parks for which additional management action is required if we want to satisfy all visitors completely. However, if managerial actions aspire to increase the positive gap value (i.e., to exceed visitors’ expectations), a more detailed study will be needed. As a starting point, we can mention that in our case, the highest positive gap value was found in the PNA with the highest protection category (Aigüestortes i Estany de Sant Maurici National Park; see Table 1), more pristine areas, and a low level of human intervention. If we were able to identify some specific features, beyond the natural characteristics of the PNA, that increased the positive gap, managers could be in a position to adapt their offers to make the parks more suitable and attractive to their target audiences where health motivations are concerned, thereby increasing health benefits.


**(Q5) For each PNA, were there any differences in the gaps between satisfaction and motivation that can be explained by sociodemographic characteristics and visit behaviors?**


In general, the differences found in our study can be explained by visitor behavior rather than by sociodemographic characteristics. For instance, only two of the sociodemographic variables (gender and health perception) differ in relation to level of satisfaction, whereas all five recreational behavior variables differ. While we do not have any specific results from other studies with which to compare our own directly, our results could relate to those of Lemieux et al. [22] or Romagosa [27], who found that women rather than men often perceived greater well-being benefits derived from visits. As it was expected according to classic studies targeting low-physical activity individuals and health outcomes [51,52], our results also show that visitors whose health perception was lower obtained indeed greater physical and social health benefits after doing physical activity in the PNA. Another important finding was the link between some kind of physical activity during the visit and higher levels of satisfaction in both the physical and the spiritual dimensions. However, more data are needed to establish any connections between sociodemographic characteristics or recreational behavior, and health and well-being satisfaction.

## 5. Study Limitations and Managerial Recommendations

The number of respondents is reliable and representative, as previously mentioned in data collection and sampling strategy section. One of our limitations was that data collection was performed during spring and early summer (April to July) and extended time of field work may identify different visitor profiles.

From our point of view, future studies should include more than one PNA according to their protection categories and even from different geographically and cultural settings, including, preferably, collecting data throughout the year.

Managerial recommendations specifically addressed to a particular PNA should first identify the assets of the PNA, to link the PNA features with specific health dimensions (e.g., spiritual health related with the PNA context) or public health policies and actions (e.g., www.eupap.org).

Finally, data obtained from such studies may be used for the development of a comprehensive monitoring, evaluation and reporting framework for the effectiveness of PNA management on health and well-being promotion.

## 6. Conclusions

Previous studies have analyzed and discussed the roles/benefits of PNAs in relation to human health and well-being. However, the real outcomes generated by these areas in terms of satisfaction (the difference between motivations and benefits) have not been examined empirically. This is the contribution of this study, which is based on the application of IPA and GAP analysis. The results obtained generally show that in our set of PNAs there is a concordance between what visitors want (motivation) and what visitors get (benefits), and consequently high visitor satisfaction levels. However, important differences between PNAs were found regarding human health and well-being motivations for visits and the expected benefits. Although these differences could be explained in part by sociodemographic characteristics and recreational behavior, looking at each area’s identity or character as perceived by the visitor opens an interesting perspective. In this case, the specialization of each park with regard to some of the specific dimensions analyzed here could be a good way to optimize resources. Finally, the results obtained indicate, even if inconclusively, a link between PNA category and level of satisfaction, a link that would merit much more detailed study.

## Figures and Tables

**Figure 1 ijerph-17-06746-f001:**
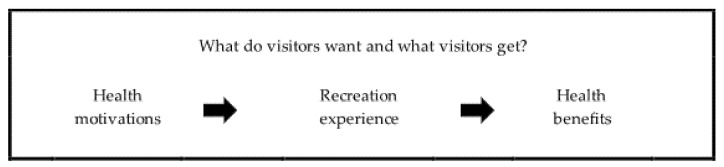
Simplified formulation of the motivations–benefit relationship. Adapted from [40].

**Figure 2 ijerph-17-06746-f002:**
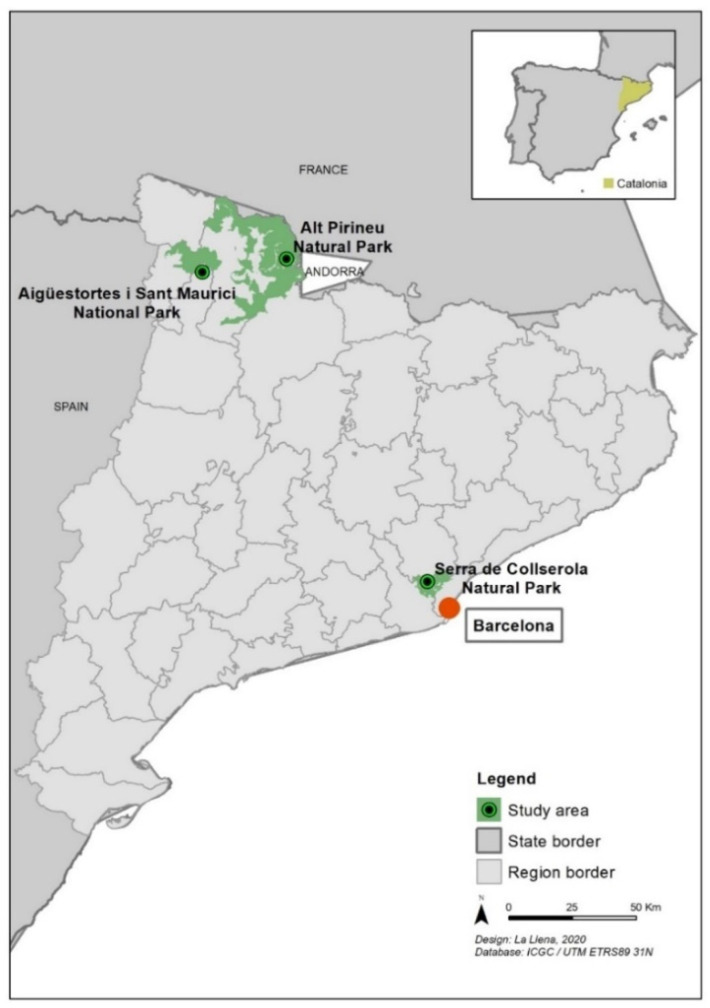
Location of the three protected natural area studies: Aigüestortes i Estany de Sant Maurici National Park, Alt Pirineu Natural Park and Serra de Collserola Periurban Park.

**Figure 3 ijerph-17-06746-f003:**
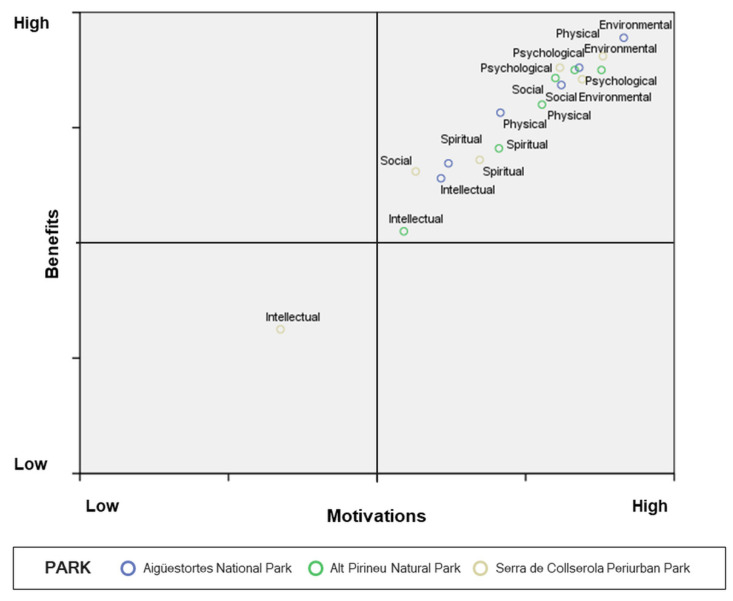
Motivation–benefits gap analysis for the three PNAs.

**Table 1 ijerph-17-06746-t001:** Main characteristics of the parks.

Features	Aigüestortes i Estany de Sant Maurici National Park (A)	Alt Pirineu Natural Park (P)	Serra de Collserola Periurban Park (C)
General characteristics			
Category	National Park EU’s Natura 2000 IUCN ^1^ category II	Natural Park EU’s Natura 2000 IUCN Category V	Periurban Park EU’s Natura 2000 IUCN Category V
Year of creation	1955–1996	2003–2018	1987
Area	40,852 ha	79,317 ha	8295 ha
Location	Alta Ribagorça, Pallars Sobirà, Pallars Jussà and Val d’Aran. Lleida Province	Pallars Sobirà and l’Alt Urgell. Lleida Province	Baix Llobregat, el Vallès Occidental and Barcelonès. Barcelona Province
Specific physical, social and recreational characteristics			
Landscape characteristics	High mountain with large number of lakes and meadows	High mountain Alpine forest	Low mountain Mediterranean forests
Human intervention	Low	Medium	High
Trails	Total length: 120 km	Total length: 175 km	Total length: 456 km
Annual visitors (2018)	400,000	345,000	5,000,000
Available public facilities	Parking and picnic areas, signposts, viewpoints, shelters	Parking and picnic areas, signposts, viewpoints, shelters	Parking and picnic areas, signposts, viewpoints. No shelters
Maps	Hiking trails	Hiking and biking trails	Hiking and biking trails
Recreational and Sporting activities	Recreational hiking, hiking, ^2^ mountaineering, trail running	Recreational hiking, hiking, mountaineering, mountain biking, trail running	Recreational hiking, hiking, mountaineering, mountain biking, trail running

^1^ International Union for Conservation of Nature. ^2^ “Recreational hiking” refers to walking at a slow pace, whereas ”hiking” refers to brisk walking, as described elsewhere [31].

**Table 2 ijerph-17-06746-t002:** Visitors’ sociodemographic characteristics and visit behavior in the three protected natural areas (PNAs) (N = 360).

Variables	Aigüestortes (*n* = 120)	Pirineu (*n* = 120)	Collserola (*n* = 120)	Total Sample	Comparison
Sociodemographics					
Gender					
Male	49.2	58.8	65.5	58.0	χ2 (df) = 6.847 (2), *p* = 0.033 *
Female	50.8	41.2	34.2	42.0	
Age					
18–25	7.1	6.8	6.0	6.6	χ2 (df) = 18.055 (8), *p* = 0.021 *
26–35	8.8	7.7	19.7	12.1	
36–45	23.9	16.2	23.1	21.0	
46–55	18.6	38.5	30.8	29.4	
55+	41.6	30.8	20.5	30.8	
Educational level					
No university degree	44.1	48.7	35.0	42.5	χ2 (df) = 4.731 (2), *p* = 0.094
University degree	55.9	51.3	65.0	57.5	
Health perception					
Very good or excellent	45.0	50.0	35.0	43.4	χ2 (df) = 2.507 (2), *p* = 0.286
Good	45.8	39.2	60.0	48.3	
Poor or fair	9.2	10.8	5.0	8.4	
Recreational Behavior					
Frquency of visit (last two years)					
First time	19.3	5.8	2.5	9.2	H (df) = 64.107 (2), *p* < 0.001 *
Second time	25.2	7.5	5.9	12.8	
3–4 times	23.5	15.0	17.6	18.7	
More than 4 times	31.9	71.7	73.4	59.2	
Group composition					
Individual	4.2	4.2	26.7	11.7	χ2 (df) = 44.915 (6), *p* < 0.001 *
Partners	34.2	25.4	19.2	26.3	
Family	23.3	20.3	13.3	19.0	
Group of friends	38.3	50.1	40.8	43.0	
Duration of visit					
Half day	80.7	55.0	99.2	78.3	χ2 (df) = 77.117 (4), *p* < 0.001 *
One day	18.5	31.7	0.8	17.0	
More than one day	0.8	13.3	0.0	4.7	
Recreational activity					χ2 (df) = 156.905 (10), *p* < 0.001 *
Remain at entrance	5.9	33.3	2.5	14.0	
Recreational hiking (slow walking)	50.8	20.0	28.3	33.0	
Hiking (brisk walking)	36.4	25.0	21.7	27.7	
Mountain biking	0.8	14.2	21.7	12.3	
Trail running	1.7	0.0	25.8	9.2	
Other activities	4.2	7.5	0.0	3.9	

Values are percentages of the number of individuals for each park. * Significant differences (Chi-square or Kruskal–Wallis test).

**Table 3 ijerph-17-06746-t003:** Rating of health and well-being motivations for visiting the PNA. Descriptive statistics and tests of significance (N = 360).

Motivations	1 *n* (%)	2 *n* (%)	3 *n* (%)	4 *n* (%)	5 *n* (%)	Sum	Mean	SD
Environmental	6 (2.2%)	6 (1.7%)	24 (6.7%)	76 (21.1%)	246 (68.3%)	89.4%	4.52	0.867
Psychological	11 (3.1%)	12 (3.3%)	41 (11.4%)	87 (24.2%)	209 (58.1%)	82.3%	4.31	1.005
Physical	12 (3.3%)	24 (6.7%)	49 (13.6%)	88 (24.4%)	187 (51.9%)	76.3%	4.15	1.097
Social	45 (1.5%)	23 (6.4%)	36 (10.3%)	75 (20.8%)	191 (50.3%)	71.1%	3.90	1.407
Spiritual	44 (12.2%)	32 (8.9%)	57 (15.8%)	96 (26.7%)	131 (36.4%)	63.1%	3.66	1.367
Intellectual	68 (18.9%)	70 (19.4%)	85 (23.6%)	74 (20.6%)	63 (17.5%)	38.1%	2.98	1.364

Sum corresponds to the sum of values 4 and 5.

**Table 4 ijerph-17-06746-t004:** Rating of perceived health and well-being benefits derived from visiting the PNA. Descriptive statistics and tests of significance (N = 360).

Benefits	1 *n* (%)	2 *n* (%)	3 *n* (%)	4 *n* (%)	5 *n* (%)	Sum	Mean	SD
Environmental	6 (1.7%)	8 (2.2%)	18 (5.0%)	72 (20.0%)	256 (71.1%)	91.1%	4.57	0.825
Psychological	6 (1.7%)	8 (2.2%)	18 (5.0%)	92 (25.6%)	236 (65.6%)	91.1%	4.51	0.828
Physical	10 (2.8%)	18 (5.0%)	32 (8.9%)	89 (24.7%)	211 (58.6%)	83.3%	4.31	1.015
Social	36 (10.0%)	16 (4.4%)	24 (6.7%)	71 (19.7%)	213 (59.2%)	78.9%	4.14	1.312
Spiritual	38 (10.6%)	30 (8.3%)	59 (16.4%)	93 (25.8%)	140 (38.9%)	64.7%	3.74	1.332
Intellectual	77 (21.4%)	62 (17.2%)	83 (23.1%)	71 (19.7%)	67 (18.6%)	38.3%	2.98	1.364

Sum corresponds to the sum of values 4 and 5.

**Table 5 ijerph-17-06746-t005:** Visitors’ health and well-being motivations. Descriptive statistics and comparisons between parks.

Motivations	Aigüestortes M (SD)	Alt Pirineu M (SD)	Collserola M (SD)	Kruskal–Wallis H, *p*
Environmental	4.66 (0.716)	4.51 (0.810)	4.38 (1.030)	5.944, *p* = 0.051
Psychological	4.36 (0.877)	4.33 (1.072)	4.23 (1.059)	1.213, *p* = 0.545
Physical	3.83 (1.248) ^C^	4.11 (1.083) ^C^	4.52 (0.809) ^AP^	22.278, *p* < 0.001
Social	4.24 (1.264) ^C^	4.20 (1.127) ^C^	3.26 (1.574) ^AP^	36.269, *p* < 0.001
Spiritual	3.48 (1.390) ^p^	3.82 (1.316)	3.69 (1.383)	4.099, *p* = 0.129
Intellectual	3.43 (1.268) ^C^	3.18 (1.281) ^C^	2.35 (1.313) ^AP^	40.052, *p* < 0.001

A = Post hoc statistically significant with Aigüestortes. P = Post hoc statistically significant with Alt Pirineu. C = Post hoc statistically significant with Collserola. The level of significance for the U Mann–Whitney tests was adjusted to *p* < 0.017 (*p* = 0.05/3).

**Table 6 ijerph-17-06746-t006:** Visitors’ perceived health and well-being benefits. Descriptive statistics and comparisons between parks.

Benefits	Aigüestortes M (SD)	Alt Pirineu M (SD)	Collserola M (SD)	Kruskal–Wallis H, *p*
Environmental	4.78 (0.542) ^PC^	4.50 (0.850) ^A^	4.43 (0.984) ^A^	12.148, *p* = 0.002
Psychological	4.52 (0.860)	4.50 (0.840)	4.52 (0.788)	0.282, *p* = 0.868
Physical	4.13 (1.104) ^C^	4.20 (1.097) ^C^	4.62 (0.735) ^AP^	16.229, *p* < 0.001
Social	4.37 (1.181) ^C^	4.43 (0.876) ^C^	3.62 (1.620) ^AP^	21.846, *p* < 0.001
Spiritual	3.69 (1.321)	3.82 (1.341)	3.72 (1.342)	0.852, *p* = 0.653
Intellectual	3.56 (1.235) ^PC^	3.10 (1.293) ^AC^	2.25 (1.367) ^AP^	52.800, *p* < 0.001

A = Post hoc statistically significant with Aigüestortes. P = Post hoc statistically significant with Alt Pirineu. C = Post hoc statistically significant with Collserola. The level of significance for the U Mann–Whitney tests was adjusted to *p* < 0.017 (*p* = 0.05/3).

**Table 7 ijerph-17-06746-t007:** Results from the gap analysis.

Parks	Aigüestortes			Alt Pirineu			Collserola			H, *p*
**Dimensions**	Mot.	Ben.	Gap	Mot.	Ben.	Gap	Mot.	Ben.	Gap	Kruskal–Wallis
Environmental	4.66	4.78	0.12	4.51	4.50	−0.01	4.38	4.42	0.04	3.498–0.174
Psychological	4.36	4.52	0.16	4.33	4.50	0.17	4.23	4.52	0.27	3.410–0.182
Physical	3.83	4.13	0.30 ^PC^	4.11	4.20	0.09 ^A^	4.52	4.62	0.10 ^A^	11.178–0.004
Social	4.24	4.37	0.13	4.20	4.43	0.23	3.26	3.62	0.36	2.567–0.277
Spiritual	3.48	3.69	0.21 ^C^	3.82	3.82	0.00	3.69	3.72	0.03	6.272–0.043
Intellectual	3.43	3.56	0.13	3.18	3.10	−0.08	2.35	2.25	−0.10	4.642–0.098

Note: Mot. = Motivation, Ben. = Benefits. A = Post hoc statistically significant with Aigüestortes. P = Post hoc statistically significant with Alt Pirineu. C = Post hoc statistically significant with Collserola. The level of significance for the U Mann–Whitney tests was adjusted to *p* < 0.017 (*p* = 0.05/3).

**Table 8 ijerph-17-06746-t008:** Descriptive statistics and test of significance for results of GAP analysis dimensions.

Dimensions	Gap	Age ^1^	Gender	Health Perception	Educational Level ^2^	Frequency of Visit ^1^	Group Composition ^1^	Duration of Visit ^1^	Activity ^1^
Aigüestortes									
Environmental	0.12	0.153	0.342	0.405	0.987	0.042	0.007	0.843	0.000 **
Psychological	0.16	0.252	0.300	0.502	0.923	0.028	0.446	0.269	0.281
Physical	0.30	0.406	0.029 *	0.013 *	0.403	0.002 *	0.065	0.204	0.021 *
Social	0.13	0.885	0.835	0.044 *	0.657	0.732	0.449	0.682	0.047 *
Spiritual	0.21	0.432	0.002 *	0.356	0.941	0.204	0.242	0.809	0.081
Intellectual	0.13	0.355	0.252	0.334	0.829	0.762	0.866	0.700	0.476
Alt Pirineu									
Environmental	−0.01	0.591	0.708	0.906	0.939	0.371	0.283	0.672	0.677
Psychological	0.17	0.390	0.220	0.597	0.077	0.074	0.844	0.016 *	0.120
Physical	0.09	0.512	0.984	0.570	0.970	0.186	0.387	0.234	0.449
Social	0.23	0.742	0.785	0.983	0.586	0.505	0.703	0.304	0.208
Spiritual	0.00	0.439	0.658	0.823	0.671	0.965	0.087	0.513	0.262
Intellectual	−0.08	0.435	0.466	0.373	0.436	0.819	0.904	0.070	0.654
Collserola									
Environmental	0.04	0.053	0.813	0.950	0.930	0.250	0.766	0.047	0.527
Psychological	0.27	0.618	0.359	0.049	0.908	0.497	0.396	0.063	0.200
Physical	0.10	0.985	0.222	0.050	0.488	0.767	0.365	0.884	0.236
Social	0.36	0.249	0.076	0.348	0.757	0.888	0.008 *	0.721	0.394
Spiritual	0.03	0.256	0.937	0.246	0.788	0.688	0.273	0.972	0.006 *
Intellectual	−0.10	0.215	0.687	0.120	0.757	0.462	0.460	0.902	0.185

^1^ Mann–Whitney (variables with two categories); ^2^ Kruskal–Wallis (more than two categories). Values included in the table are the *p* values corresponding to these analyses. * *p* < 0.05; ** *p* < 0.001.

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
