# Peer review of "Health and Well-Being in Protected Natural Areas—Visitors’ Satisfaction in Three Different Protected Natural Area Categories in Catalonia, Spain"

_ijerph, 2020, doi:10.3390/ijerph17186746_

Round 1
Reviewer 1 Report
An interesting study indeed; methodological approach and results are well explained but could profit from a some additional clarification respectively discussion.
First of all, the composition of the samples is presented in comparison with the catalan population, which is fine of course. The reader is left questionning, however, whether the collected sub-samples are indeed representative of catalan PNAs and in particular for the specific parks when reading section 2. Do you have any comparative visitor analyses/studies (e.g. the ones mentioned in section 4, at the beginning) about visitor demographics to discuss them already in the methodological section? Additionally, the paper might merit acknowledging the uncertainities regarding the representativeness as the sub-samples are also not too extensive in total number and regarding their collection process.
Results are very well presented. Some aspects woulf profict from elaborationg further the influence of variables such as e.g. the duration, the frequency/habituation to the areas and the offers in the areas. For instance you address the intellectual motivitations in the sub-urban park in 3.5 and also in section 2, which are low and suggest attention by the management on these aspects. It would be interesting to discuss the likelihood that frequent return to this area and conjoint purpose of "escaping daily life" might be a contrapoint to intellectual activities per se, while physical and psychological/emotional aspects are highly important. Additionally, I was wondering how far the diverse natural structures might have influenced e.g. the spiritual variables. Is there any option to analyse or at least adress this in the discussion (you mention the natural conditions in the beginning). In 3.6. some possible correlations between variables are discussed, which is a an important starting point to analyse impacting factors also with relevance for managament activities. To amplify the outreach of the publication, could you imagine to discuss some more management relevant findings as well as transferability and/or limitations of them to international context?
Reviewer 2 Report
This study has very interesting results that help in planning an effective PNA management direction by finding the difference between health and well-being motivation and satisfaction according to PNA. In addition, since the desired motives of visitors found in this study are related to the expected benefits, it will be helpful in establishing a park management plan that optimizes resources according to the characteristics of each park in the future. However, in this study, the reliability of the results is insufficient because only one park is selected for each of the three parks with different characteristics. Therefore, the authors will be able to increase the credibility of the results of this study by investigating more parks by type of natural park, national park and peri-urban park in future studies.
Please modify the following minor details.
- 2.2. Data collection : Please specify if the survey respondents received any compensation.
Table 8 : Mark a significant p value with an *.- Since this study derives the results from the visitors' questionnaire responses, it is necessary to mention the individual health conditions that can affect the visitor's motivation. If the authors have investigated the individual health status of the visitors, the results are presented in Table 2, otherwise the authors should address the limitations of the research in this regard in the Q5 section of the disccussion.
- It is suggested that the authors can increase the reliability of research results by investigating more parks by park type in future research. Please mention this in conclusion as a future study.
